# Relationship between the skeletal muscle mass index and physical activity of Japanese children: A cross-sectional, observational study

**Tadashi Ito** [1,2]*, **Hideshi Sugiura**[2], **Yuji Ito** [3], **Koji Noritake**[4], **Nobuhiko Ochi**[3]

1 Three-Dimensional Motion Analysis Room, Aichi Prefectural Mikawa Aoitori Medical and Rehabilitation Center for Developmental Disabilities, Okazaki, Japan, 2 Department of Physical Therapy, Graduate School of Medicine, Nagoya University, Nagoya, Japan, 3 Department of Pediatrics, Aichi Prefectural Mikawa Aoitori Medical and Rehabilitation Center for Developmental Disabilities, Okazaki, Japan, 4 Department of Orthopedic Surgery, Aichi Prefectural Mikawa Aoitori Medical and Rehabilitation Center for Developmental Disabilities, Okazaki, Japan

* sanjigen@mikawa-aoitori.jp

**Data Availability Statement:** All relevant data are within the paper and its Supporting Information files.

## Abstract

Regular physical activity is an important component of physical health of children and has been associated with increasing skeletal muscle mass and muscle strength. Children with low levels of physical activity may experience health problems, such as loss of muscle mass, later in life. Thus, it may be valuable to identify declining physical function in children who do not perform the recommended amount of physical activity. Therefore, we aimed to evaluate the relationship between the amount of physical activity performed for $\geq$60 min per day for $\geq$5 days per week and the skeletal muscle mass index and physical function in young children. In total, 340 typically developing children aged 6–12 years (175 girls; average age, 9.5±1.9 years) were included in this cross-sectional study. We evaluated the proportion of children performing the recommended minimum of 60 min of daily moderate-to-vigorous physical activity at least 5 days per week. The skeletal muscle mass and Gait Deviation Index scores, gait speed, grip strength, Five Times Sit-to-Stand test results, Timed Up-and-Go test results, one-leg standing time, and gait efficiency were evaluated. Multiple logistic regression analyses were performed to assess the association of moderate-to-vigorous physical activity with the skeletal muscle mass index, percent body fat, and physical function, after controlling for confounding factors (age and sex). A logistic regression analysis revealed that the skeletal muscle mass index was independently associated with moderate-to-vigorous physical activity (odds ratio, 2.34; 95% confidence interval, 1.17–4.71; $P$ = 0.017). Performance of moderate-to-vigorous physical activity for $\geq$5 days per week for $\geq$60 min per day was associated with the skeletal muscle mass index score of Japanese children. Our findings highlighted the importance of performing moderate-to-vigorous physical activity for the development of skeletal muscle mass in children.

**Funding:** The authors received no specific funding for this work.

**Competing interests:** The authors have declared that no competing interests exist.

## Introduction

A physically active lifestyle improves muscle health or physical health and has a positive effect on body composition and physical function in childhood and adolescence [1]. Therefore, regular physical activity is important for the physical health of children [2,3]. International recommendations by the World Health Organization, 2018 United States guidelines, and United Kingdom Chief Medical Officers indicate that children and adolescents should perform moderate-to-vigorous physical activity (MVPA) for ≥60 min per day for >5 days per week [1,4,5]. Physical activity varies according to demographics and social factors, the most notable of which are ethnicity and sex [6]. In 2008, the global proportion of adolescents aged 13–15 years who did not perform the recommended MVPA was 80.3% [7]. In 2010, these rates among children and adolescents aged 11–17 years were 78.4% and 84.4% for boys and girls, respectively [8]. In 2016, a study on children aged 9–11 years from 12 countries revealed that 44.1% of them performed at the recommended MVPA levels [9]. In 2019, similar findings were reported in a study on healthy children aged 4–13 years in Australia (42.0%) [10]. Recently, a study in Japan showed that 74.7% of boys and 55.2% of girls from a sample of 657 children aged 9–15 years fulfilled the physical activity guidelines of the Japan Sports Association (≥7 h per week) [11]. However, these studies did not assess the relationship between the recommended levels of MVPA and physical function (e.g., muscle strength, balance function, and gait quality). Nonetheless, whether the physical function of children is prone to decline when they do not perform the recommended MVPA remains unclear.

Physical function refers to the ability to perform physical movements and includes grip strength, an indicator of total muscle strength; gait quality, speed, and efficiency, indicators of gait ability; the Five Times Sit-to-Stand Test (FTSST), an indicator of functional performance; the Timed Up-and-Go test (TUG), an indicator of mobility; and the one-leg standing time (OLST), an indicator of postural and balance control. Measuring grip strength may help track changes in children's health [12,13]. Gait speed and quality are critical elements of physical function and have been reported to serve as fundamental indicators of gait development in children [14–16]. The physiological cost index of gait has been widely used to assessment gait efficiency in healthy children [17]. A previous study reported that the FTSST and TUG are feasible and reliable tools for both children exhibiting typical development and those with disabilities [18–22]. The OLST has been proposed as a fundamental indicator of balance function in children [23,24].

Colley et al. reported that walking ability is an environmental factor that may influence physical activity [25]. Additionally, physical activity has been reported to be important for increasing the skeletal muscle mass and muscle strength of children [26–29]. More recently, research has shown that children may develop sarcopenia and chronic diseases [30]. Children with low MVPA levels may experience health problems, such as sarcopenia, later in life. Therefore, it may be valuable to identify signs of declining physical function and skeletal muscle mass in children who do not perform the recommended levels of MVPA. However, not all studies have consistently confirmed the associations among the skeletal muscle mass, physical function, and physical activity of children. Furthermore, to the best of our knowledge, no study has investigated these relationships among school-age children in Asia. Therefore, in this study, we examined the associations among the skeletal muscle mass, physical function, and MVPA of Japanese school-age children.

## Materials and methods

### Study population

Of the 48 elementary schools in Okazaki City, two schools introduced to us by the Okazaki City Board of Education, from which we received informed consent for study participation,

were included. All 1997 students attending these two elementary schools in Okazaki City were invited to Aichi Prefectural Mikawa Aoitori Medical and Rehabilitation Center for Developmental Disabilities to undergo a medical examination and physical function evaluation. In total, 400 school-age children (6–12 years) were recruited for this cross-sectional study between January 2018 and December 2019. The exclusion criteria were as follows: orthopedic, neurological, ophthalmologic, auditory, respiratory, or cardiovascular abnormalities that could affect the results of physical function tests; inability to complete the physical function tests; substandard scores for the Raven's Colored Progressive Matrices and the Picture Vocabulary Test-Revised, which indicate intellectual disability [31,32]; and previously diagnosed autism spectrum or attention-deficit hyperactivity disorder. Of the 400 potentially eligible children, 60 were excluded, and finally, 340 participants were enrolled in this study (Fig 1). Of note, six of the excluded participants were being treated with medications that could influence body composition and body weight. None of the other participants were receiving such medications. The study was conducted in accordance with the Declaration of Helsinki, and the protocol was approved by the Ethics Committee of the Aichi Prefectural Mikawa Aoitori Ethics Review Board (approval number, 29002). The manuscript was prepared according to the Strengthening the Reporting of Observational Studies in Epidemiology (STROBE) guidelines. The legal guardians of all participants provided written informed consent for the children's participation in the study and for the publication of any identifying information. In addition, all the children assented to study participation.

## Measurement of variables

**Moderate-to-vigorous physical activity (MVPA) questionnaire.** The Japanese version of the World Health Organization Health Behavior in School-age Children was used to evaluate the MVPA levels [33]. Participants provided self-responses to the physical activity questionnaire. The questionnaire was administered to individuals who performed ≥60 min MVPA per day for ≥5 days per week, which is the recommended MVPA levels.

**Assessment of appendicular skeletal muscle mass.** A multi-frequency bioelectrical impedance analyzer (MC-780; Tanita, Tokyo, Japan) was used to measure the appendicular skeletal muscle mass and percent body fat. Standard positioning was used to perform measurements, as described in the instruction manual. The participants stood with the soles of their feet in contact with the anterior and posterior foot electrodes and grabbed the hand electrodes to maintain contact with the palm and thumb of each hand. Arms were extended and hung down in a natural standing position, and skin-to-skin contact was avoided. Measurements were performed by well-trained physical therapists or research assistants and completed within 15 s. The bioelectrical impedance analyzer resistance was obtained at three electrical frequencies: 5, 50, and 250 kHz. The skeletal muscle mass index (SMI) score was calculated as the appendicular skeletal muscle mass divided by height squared. The percent body fat of participants was calculated using the bioelectrical impedance analyzer. The multi-frequency bioelectrical impedance analyzer is used to determine the relationship between the measured volume of an electrical resistance and the conductor. The analyzer uses multiple frequencies to differentiate intracellular fluid from extracellular fluid and, therefore, provides an estimation of total body water [34]. Fat and bone have correspondingly high impedance and low conductivity, and skeletal muscle is an electrolyte rich tissue with low resistance. The analyzer has been demonstrated to be non-invasive, convenient, and economic and has been used to evaluate muscle development and muscle cross-sectional areas of children [35,36]. Bioelectrical impedance analysis (BIA) was only performed 2 h after meals, as described in the instruction manual.

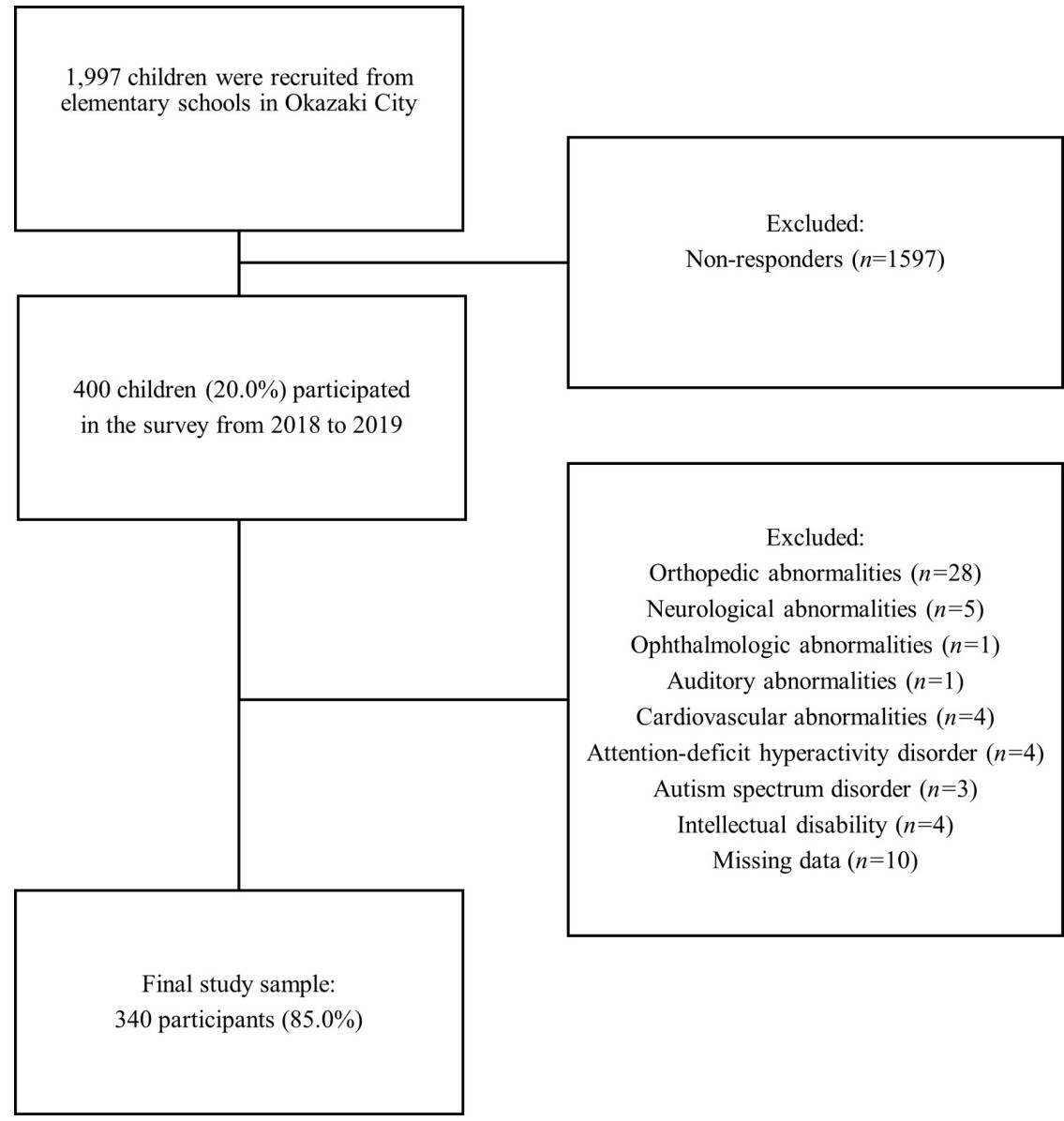

**Fig 1. Flowchart of the enrollment of study participants.**

**Gait analysis.**   Measurements were performed using an eight-camera motion analysis system with a sampling frequency of 100 Hz (type MX-T 20S; Vicon, Oxford, UK). The Plug-In-Gait lower body Ai marker set (Vicon) was positioned on the anterior and posterior superior iliac spines, lateral femoral, lateral part of the knee joint, lateral lower limb, lateral malleoli of the ankle joint, heads of the second metatarsal, and both sides of the calcaneus bones by a physiotherapist with 10 years of experience in clinical gait analysis. An instrumented three-dimensional gait analysis was performed using the Plug-In-Gait (Vicon) to determine the kinematics of the lower extremities at the pelvis, hips, knees, and both sides of the ankles. The participants were recorded as they walked barefoot at a self-controlled speed on 8-AMTI OPT force plates (Advanced Mechanical Technology, Inc., Watertown, MA, USA) during three trials [37,38]. The Gait Deviation Index (GDI) of the participants was used to detect the gait

pattern and was calculated using the Vicon current pipeline, as described in the study by Schwartz and Rozumalski [16]. The mean GDI score was calculated using the results of three gait trials to analyze the right and left lower limbs [16]. The GDI is an important tool that represents the overall gait pattern using numerical values; it is determined based on a score derived from the kinematic data points of the three-dimensional gait analysis of the pelvis, hip, knee, and ankle in the sagittal plane; the pelvis and hip in the frontal plane; and the pelvis, hip, and foot progression in the horizontal plane [16]. The GDI has been demonstrated to have high validity and excellent intra-rater and inter-trial reliability [39,40]. The normal gait speed results were recorded in m/s [15]. The mean normal gait speed was calculated using the results of three gait trials analyzing the right and left lower limbs.

**Grip strength.** Grip strength was measured (in kg) as the participant's mean grip strength of both hands using an adjustable Smedley-type handheld dynamometer (GRIP-D; Takei Ltd., Niigata, Japan). The dominant hand was measured first, followed by the non-dominant hand. Grip strength of both hands was measured once each with the participants in the sitting position with their shoulder abducted and neutrally rotated, the elbow at extension, and the forearm and wrist in the neutral position [41]. The participants were instructed to squeeze the handle of the dynamometer as hard as they could and sustain for 5 s [42]. Verbal encouragement (i.e., "Keep it up, keep it up") was provided during the test [42]. Prior to grip strength assessment by an experienced physical therapist, the participants were allowed to perform one practice trial as a learning attempt. The reliability of grip strength measurement was reported to be excellent in a full elbow-extension position [43].

**Five Times Sit-To-Stand test (FTSST).** The FTSST, which is a functional performance measurement test, was used to evaluate lower muscle strength. A bench chair, which was height-adjusted to 90˚ of knee flexion, was used for the test. The participants sat on a chair with their arms crossed over the chest. Next, they were asked to stand up and sit down five times as quickly as possible without using arm support [18,19,44]. The FTSST began with the word "go" and stopped when the participant stood up after the fifth repetition; the time duration was recorded using a stopwatch (s). The participants were allowed to perform one practice trial before the actual test [18,19,44].

**Timed UP-and-Go test (TUG).** The TUG test was used to evaluate functional mobility and the dynamic balance function. To perform the TUG test, the participants were asked to sit on an adjusted chair with 90˚ knee flexion. This test involved rising from an armchair, walking 3 m at a usual speed, turning around, walking back to the chair, and sitting down [20–22,45,46]. These instructions were clearly communicated to participants. The TUG test score (in s) after one trial was used for the analysis. A demonstration was provided, and one practice trial was allowed for each participant.

**One-leg standing time (OLST).** The OLST was used to determine how long a participant could remain standing on one leg with their eyes open, which indicates the static balance function [47–49]. The arms had to be held alongside the body. The participants were asked to lift one foot from the floor, making sure not to brace the lifted leg against the support leg. The test ended when the lifted leg touched the support leg, when the lifted leg touched the floor, or after 120 s of maintaining successful balance. The mean OLST was calculated using the mean OLST (in s) of the right and left lower limbs. OLST was reported to have a good test-retest reproducibility and inter-rater reliability [23,24,50].

**Gait efficiency.** During the 2-min walking distance test, the participants were instructed to walk at a comfortable pace and without further encouragement on a flat and hard surface for 2 min to determine how far they could walk during that time [51,52]. The 2-min walking distance test was conducted following a standardized protocol [53] along a 10-m straight corridor. The examiner walked behind the participant and provided clear instructions; the

participants were allowed to stop and rest. Gait efficiency during the 2-min walking distance test was measured using the POLAR M 600 (Polar Electro, Espoo, Finland), a sport-optimized smartwatch designed to measure the heart rate. The smartwatch was held in place by a wrist-band on the right wrist of the participant. Before gait efficiency assessment, each participant was required to sit quietly for a 3-min rest period. Gait efficiency was calculated based on net efficiency as follows:

Physiological cost index = (gait heart rate − resting heart rate)/2-min walking distance average gait speed (beats/m/s) [54,55].

## Sample size

The sample size was calculated using G*Power version 3.1.9.4 (Heinrich Heine University, Düsseldorf, Germany) [56,57]. Additionally, the number of children who performed MVPA ≥5 days per week for ≥60 min per day was used to calculate the sample size. Based on the assumption that 42.0% of healthy children performed the recommended physical activity levels [10], we calculated an odds ratio (OR) of 1.91. A power analysis was conducted for the logistic regression to determine the sufficient sample size using an alpha level of 0.05, power level of 0.8, OR of 1.91, and a two-tailed test. Based on the aforementioned assumption, the desired sample size was 308.

## Statistical analysis

When appropriate, *t*-tests and chi-square tests were used to compare the physical function of children who performed MVPA ≥5 days per week at ≥60 min per day (defined as group 1) and that of children who performed MVPA <5 days (defined as group 2). A Bonferroni correction was used to compare the physical characteristics of groups 1 and 2, with an α threshold of 0.0125 (0.05/4). Moreover, a Bonferroni correction was used to compare the physical function of these groups, with an α threshold of 0.0056 (0.05/9). The effect sizes were calculated using *r* and the Cramer's *V* test (sex), which were considered small, moderate, and large when *r* = 0.1 or -0.1, *r* = 0.3 or -0.3, and *r* = 0.5 or -0.5, respectively. We confirmed that the children showed no significant sex-based differences in their MVPA values (*P* = 0.109; effect size [r], -0.1) (S1 Table). Moreover, our results revealed that seasonal variations showed no significant associations with MVPA (*P* = 0.061; $\eta^2$ = 0.02) (S2 Table). Participant data are expressed as means and standard deviations or median values (ranges), and were compared using the independent t-test or Mann–Whitney U-test. Spearman's rank correlation analysis was performed to determine the relationship between the SMI score and percent body fat; it was confirmed that the correlation was not high (*r* = 0.454; *P*<0.0001) before the logistic regression analysis was performed. The logistic regression analysis was conducted to examine the associations among the SMI score, percent body fat, physical function, and MVPA. *P*<0.05 was considered statistically significant. A multivariate logistic regression model was used to determine the OR for the SMI score, percent body fat, GDI, gait speed, grip strength, FTSST, TUG test, OLST, gait efficiency, after controlling for confounding factors (age and sex) in group 1. Data management and statistical computations were performed using the IBM SPSS Statistics 24.0 software package (SPSS Inc., Armonk, NY, USA).

## Results

The results indicated that 82 of 165 boys (49.7%) and 105 of 175 girls (60.0%) did not perform the recommended MVPA at the time of assessment. The percentages of children who performed the recommended MVPA levels stratified according to different age groups were as follows: age 6 years, 11.8%; age 7 years, 25.0%; age 8 years, 31.6%; age 9 years, 37.7%; age 10

**Table 1. Demographic characteristics of study participants.**

| Variables | Children performing the recommended MVPA levels (*n* = 153) | Children performing substandard MVPA levels (*n* = 187) | *P* value* | Effect size (*r* or Cramer's *V* test) |
|---|---|---|---|---|
| **Age (years)[a]** | 10.0 (6.0–12.0) | 9.0 (6.0–12.0) | 0.0001 | -0.3 |
| **Height (cm)[a]** | 140.9 (110.8–164.2) | 130.7 (106.5–163.6) | 0.0001 | -0.3 |
| **Weight (kg)[a]** | 31.4 (17.8–59.6) | 27.2 (16.1–74.4) | 0.0001 | -0.2 |
| **Body mass index (kg/m²)[a]** | 16.01 (12.91–24.21) | 15.68 (12.32–29.6) | 0.069 | -0.1 |
| **Sex: Female/Male[b]** | 70/83 | 105/82 | 0.056 | 0.1 |

[a]Data are presented as medians (ranges).

[b]Differences in the proportion of sexes were derived using the chi-square test.

*P* values for age, height, weight, and body mass index were derived using the Mann–Whitney *U* test.

MVPA, moderate-to-vigorous physical activity.

years, 57.6%; age 11 years, 72.3%; and age 12 years, 53.0%. The participants' characteristics and a comparison of children who performed and did not achieve the recommended MVPA levels are summarized in Table 1.

Children who performed the recommended MVPA levels (group 1) were older (10.0 vs. 9.0 years; *P*<0.001), taller (140.9 vs. 130.7 cm; *P*<0.001), heavier (31.4 vs. 27.2 kg; *P*<0.0001) and had a higher SMI score (5.98 vs. 5.59; *P*<0.001), higher GDI score (95.76 vs. 93.38; *P* = 0.003), stronger grip strength (14.15 vs. 12.0; *P*<0.001), faster TUG time (7.29 vs. 7.62 s; *P*<0.001), and longer OLST (120.0 vs. 93.24 s; *P*<0.001) than those with substandard MVPA levels (group 2) (Table 2).

For the logistic regression analysis, 340 participants were selected after undergoing medical examination to evaluate physical function. There was a significant relationship between the MVPA and SMI score (OR, 2.34; 95% confidence interval [CI], 1.17–4.71; *P* = 0.017) (Table 3). Regarding the relationship between MVPA and percent body fat, group 1 had significantly lower OR values (OR, 0.93; 95% CI, 0.88–0.98) than group 2 (Table 3), indicating a significant

**Table 2. Physical function outcomes of study participants.**

| Variables | Children performing the recommended MVPA levels (*n* = 153) | Children performing substandard MVPA levels (*n* = 187) | *P* value* | Effect size (*r*) |
|---|---|---|---|---|
| **Skeletal muscle mass index (kg/m²)** | 5.98 (4.75–7.64)[a] | 5.59 (4.55–8.48) | 0.0001 | -0.3 |
| **Percent body fat (%)** | 12.2 (5.4–34.8)[a] | 13.2 (5.0–46.4) | 0.241 | -0.1 |
| **Gait Deviation Index (points)** | 95.76±7.40[a] | 93.38±7.10 | 0.003 | 0.2 |
| **Gait speed (m/s)** | 1.20±0.17[a] | 1.16±0.16 | 0.032 | 0.1 |
| **Grip strength (kg)** | 14.15 (6.15–26.4)[a] | 12.0 (5.75–33.0) | 0.0001 | -0.3 |
| **Five Times Sit-To-Stand test (s)** | 5.94 (3.42–11.35)[a] | 6.1 (3.28–10.6) | 0.558 | -0.03 |
| **Timed Up-and-Go test (s)** | 7.29 (4.74–10.21)[a] | 7.62 (4.57–11.57) | 0.001 | -0.2 |
| **One-leg standing time (s)** | 120.0 (20.92–120.0)[a] | 93.24 (2.64–120.0) | 0.0001 | -0.3 |
| **Physiological cost index (beat/m/s)** | 22.0 (0–57.61)[a] | 23.46 (0–76.29) | 0.446 | -0.04 |

[a]Data are presented as means±standard deviations or medians (ranges).

*P* values for the Gait Deviation Index score and gait speed were derived using an independent *t*-test. For all other variables, *P* values were derived using the Mann–Whitney *U* test.

MVPA, moderate-to-vigorous physical activity.

**Table 3. Relationship between MVPA and physical function.**

| | Children performing recommended (*n* = 153) and substandard MVPA levels (*n* = 187) | | | |
|---|---|---|---|---|
| | **B** | **SE** | **Odds ratio (95% CI)** | ***P* value** |
| **Age** | 0.18 | 0.12 | 1.2 (0.95−1.5) | 0.13 |
| **Sex** | 0.25 | 0.31 | 1.28 (0.7−2.3) | 0.428 |
| **Skeletal muscle mass index**[a] | 0.85 | 0.36 | 2.34 (1.17−4.71)[c] | 0.017 |
| **Percent body fat**[a] | -0.08 | 0.03 | 0.93 (0.88−0.98)[b] | 0.006 |
| **Gait Deviation Index**[a] | 0.04 | 0.02 | 1.04 (1.0−1.08)[b] | 0.032 |
| **Gait speed** | -1.12 | 0.92 | 0.33 (0.05−1.96) | 0.22 |
| **Grip strength** | -0.02 | 0.05 | 0.98 (0.9−1.07) | 0.678 |
| **Five Times Sit-to-Stand Test** | 0.03 | 0.1 | 1.03 (0.85−1.25) | 0.743 |
| **Timed Up-and-Go test**[a] | -0.39 | 0.13 | 0.68 (0.53−0.87)[b] | 0.002 |
| **One-leg standing time**[a] | 0.02 | 0.01 | 1.02 (1.01−1.03)[b] | 0.0001 |
| **Physiological cost index** | -0.01 | 0.01 | 0.99 (0.97−1.01) | 0.241 |

[a]Logistic regression analyses revealed that skeletal muscle mass index, percent body fat, Gait Deviation Index score, Timed Up-and-Go test results, and one-leg standing time were independently associated with MVPA.

[b]Variables with a significant odds ratio.

[c]Variables with a significantly high odds ratio.

Hosmer–Lemeshow $\chi^2$ = 5.424, *P* = 0.711.

Nagelkerke $R^2$ = 0.3.

B, partial regression coefficient; CI, confidence interval; MVPA, moderate-to-vigorous physical activity; SE, standard error.

relationship between MVPA and physical function. Children in group 1 were more likely to have higher physical function than those in group 2 (GDI: OR, 1.04 and 95% CI, 1.0–1.08; TUG test: OR, 0.68 and 95% CI, 0.53–0.87; OLST: OR, 1.02 and 95% CI, 1.01–1.03) (Table 3).

## Discussion

This study found a significant relationship between the skeletal muscle mass and substandard MVPA levels among school-aged children. Moreover, our study revealed the following important findings: 55% of children aged 6–12 years did not perform the recommended MVPA. Further, fulfillment of the recommended MVPA levels corresponded to favorable relationships with functional mobility and muscle strength (TUG), gait pattern (GDI), balance function (OLST), and percent body fat. Fulfillment of the physical activity recommendations was not associated with grip strength, FTSST, gait speed, or gait efficiency in 6–12-year-old Japanese children; the OR for skeletal muscle mass (SMI) was higher than that for gait pattern (GDI), functional mobility and balance (TUG), balance function (OLST), and percent body fat of children who performed the recommended MVPA levels. These findings highlighted the importance of performing MVPA levels to ensure the development of skeletal muscle mass in children.

Our findings may increase the awareness of the need to assess muscle mass of school-age children performing substandard MVPA levels. Although the loss of muscle mass has been associated with sarcopenia in older adults, recent evidence has shown that inactive children may also develop sarcopenia [58,59]. Sarcopenia is highly prevalent in children after liver transplantation and has been reported to be related to lower MVPA levels [28]. Moreover, a previous study revealed that disability significantly affected the muscle mass [60]. Furthermore, according to previous studies, children who are more physically active may have lower

percent body fat values and higher muscle mass values [61,62]. The findings of a Chinese study group [63] and those of Babirekere-Iriso et al. [64] indicated that physical activity may have a vital role in the development of skeletal muscle mass in children. The findings of Doaei et al. [65,66] indicated that the FTO gene may play a vital role in the increase in skeletal muscle mass and body composition in male adolescents. Furthermore, lifestyle modifications may exert their effects on obesity through changes in the expression of the FTO and IRX3 genes [65]. Moreover, lifestyle modification appeared to have an impact on obesity through changes in the expression of the FTO and IRX3 genes [67]. Therefore, performing the recommended MVPA levels should be considered an important component of the skeletal muscle mass status of children with or without a disease. From a long-term perspective, it can be assumed that low SMI values are primarily caused by low physical activity levels of children. Therefore, maintaining an optimal SMI score during childhood could improve the maximum muscle mass [68]. Moreover, the effects of exercising the musculoskeletal system seemed to have a greater impact on muscle mass during childhood than during adulthood or old age [69]. A future longitudinal study is needed to confirm this hypothesis.

There was a significant difference in the GDI between children who performed the recommended MVPA levels and those who did not. We also found a significant relationship between MVPA levels and the GDI, suggesting that children with a good gait pattern engaged in more physical activity. Gait is an important activity of daily life [70]. However, logistic regression analysis showed a small OR, indicating that GDI was significantly related to MVPA. Therefore, gait pattern was less related to MVPA than the skeletal muscle mass. However, no differences were observed in the gait speed of children who did and did not perform the recommended MVPA levels. The physiological cost index, which is considered a simple evaluation tool, has been widely used to estimate energy cost during physical activity, including gait, in various clinical scenarios [17,71–73]. However, it was not associated with MVPA in this study, indicating that children with shorter physical activity times did not necessarily exhibit greater changes in their heart rate.

This study also showed that the grip strength of children who performed a substandard MVPA levels was lower than that of those who performed the recommended MVPA levels. However, there was no significant association between grip strength and MVPA in our study, which was in line with the findings of a previous study involving preschool children [74]. Therefore, the grip strength test was found to not be the most appropriate for this population. Furthermore, there was no clear relationship between the MVPA levels and FTSST results in this study. Nonetheless, a previous study data suggested that handgrip strength correlated well with the overall muscle strength [75]. These results are of great interest because muscle strength may be independent of MVPA.

We found a significant relationship among the MVPA, TUG test, and OLST results, suggesting that children with high functional mobility and balance function engage in more physical activity. As the TUG test is a practical tool used to assess functional mobility and balance function [46,76] and because the OLST could be a high-intensity task for children [77], these tests may help evaluate the physical activity of typically developing children. However, the logistic regression analysis showed a small OR, indicating that the TUG test and OLST were significantly related to MVPA. Therefore, the aforementioned tests were deemed to be less related to MVPA levels than the degree of skeletal muscle mass.

Our analysis showed that 54.8% of school-age (6–12 years) boys and girls did not fulfill the current recommendations for daily physical activity. This finding was in contrast to the rate (79.9%) observed among adolescents aged 9–15 years in Unnan City, Japan [78]. Furthermore, a survey of Thai children aged 6–17 years ($n$ = 13,225) indicated that 76.6% of the respondents did not perform the recommended MVPA levels [79]. Another study reported that physical

activity was also associated with positive mental health in a multi-ethnic Asian population [80]. We presumed that children in our study actively participated in school clubs, and the opportunities for participation in sports activities were numerous. Differences in the MVPA levels among the children in our study and those in the studies conducted in Unnan City and in Thailand may be attributable to regional characteristics or differences in the ages of participants. However, the categorization of MVPA relied on self-report questionnaire data that could be affected by recall bias. Subjective assessment was reported to overestimate MVPA levels compared with the objective assessment [81]. Moreover, as the items for measuring preference for MVPA were not investigated for reliability and validity, a future longitudinal study is needed to confirm this hypothesis.

This study had several limitations. First, it must be emphasized that clear and causal associations could not be established in this study. Second, although participants from Okazaki City participated in this study, the generalizability of our findings is limited because the research involved only a few schools located in the city. Further studies involving all schools in Okazaki City are required in the future. Third, the questionnaire used in this study was subjective; therefore, there is a possibility that the participants overestimated their physical activity [81]. Fourth, the validity of the BIA method depends on the participant's hydration status. Limitations of BIA include assumptions involving a fixed hydration status [82]. Thus, BIA may not always be the best method to assess body composition. Finally, we could not account for the role of dietary intake, family income, and bone mineral density, which may be confounders of MVPA [26,83–85].

## Conclusions

We found that MVPA $\geq$60 min per day for $\geq$5 days per week, defined as the recommended MVPA levels, was associated with the SMI of Japanese school-age children. Moreover, the GDI score, TUG test, OLST, and percent body fat were also associated with MVPA, although less significantly. These results indicated the importance of performing the recommended MVPA levels for the development of skeletal muscle mass in children.

## Supporting information

**S1 Table. Demographic characteristics and physical function outcomes of study participants.**
(DOCX)

**S2 Table. Demographic seasonal variations in physical activity of the participants.**
(DOCX)

**S1 Database.**
(XLSX)

## Acknowledgments

We would like to thank the Okazaki City Board of Education, Okazaki City Medical Association, Aichi Pediatrics Medical Society, Yoshiki Nakanowatari, Ryohei Takasu, and Takayoshi Ono for their help recruiting the participants. We are also sincerely grateful to the staff at Naomichi Matsunaga, Yu Hamabe, Jun Mizusawa, Yoshiji Yamamoto, Yoshiki Fukaya, Yumi Aoki, Shuhei Takahashi, Daisuke Kawaguchi, Kento Iwamoto, Shota Sanada, Hiroto Takenaka, Arisa Omori, Yingzhi Gu, Yuya Shirai, Daiki Takahashi, and Aika Hishida for their assistance with data collection. We would like to thank Editage (www.editage.com) for English language editing.

## Author Contributions

**Conceptualization:** Tadashi Ito, Hideshi Sugiura.

**Data curation:** Tadashi Ito, Hideshi Sugiura, Yuji Ito.

**Formal analysis:** Tadashi Ito.

**Investigation:** Tadashi Ito, Hideshi Sugiura, Yuji Ito, Koji Noritake, Nobuhiko Ochi.

**Methodology:** Tadashi Ito, Yuji Ito.

**Project administration:** Tadashi Ito, Hideshi Sugiura, Koji Noritake, Nobuhiko Ochi.

**Software:** Tadashi Ito.

**Supervision:** Hideshi Sugiura, Koji Noritake, Nobuhiko Ochi.

**Validation:** Tadashi Ito, Yuji Ito.

**Visualization:** Tadashi Ito, Yuji Ito.

**Writing – original draft:** Tadashi Ito.

**Writing – review & editing:** Hideshi Sugiura, Yuji Ito, Koji Noritake, Nobuhiko Ochi.

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
