## [Decision Letter · Decision Letter 0]

18 Jan 2021

PONE-D-20-38059

Relationship between skeletal muscle mass index and physical activity of Japanese children: A cross-sectional, observational study

PLOS ONE

Dear Dr. Ito,

Thank you for submitting your manuscript to PLOS ONE. After careful consideration, we feel that it has merit but does not fully meet PLOS ONE’s publication criteria as it currently stands. Therefore, we invite you to submit a revised version of the manuscript that addresses the points raised during the review process.

The manuscript is well assessed by two Reviewers.

See the Reviewers' comments carefully and respond them appropriately.

We look forward to receiving your revised manuscript.

Kind regards,

Masaki Mogi

Academic Editor

PLOS ONE

Journal Requirements:

'The funders had no role in study design, data collection and analysis, decision to publish, or preparation of the manuscript.'

Reviewers' comments:

Reviewer's Responses to Questions

**Comments to the Author**

1. Is the manuscript technically sound, and do the data support the conclusions?

Reviewer #1: Yes

Reviewer #2: Yes

2. Has the statistical analysis been performed appropriately and rigorously? 

Reviewer #1: N/A

Reviewer #2: Yes

3. Have the authors made all data underlying the findings in their manuscript fully available?

Reviewer #1: Yes

Reviewer #2: Yes

4. Is the manuscript presented in an intelligible fashion and written in standard English?

Reviewer #1: Yes

Reviewer #2: Yes

5. Review Comments to the Author

Reviewer #1: Thank you for the hard-effort observational study. This study has tried to reveal the relationship between sufficient PA and skeletal muscle mass in growing children. And even more, authors tried to figure out the real physical performance by examining several time-consuming studies such as TUGT, one-leg standing time, sit to stand test results, grip strength, gait, etc.

Method

Please define when skeletal muscle mass assessment with BIA was done as the previous food, or liquid consumption could influence the results.

Would you please describe details about the grip strength? Did you have a practice trial or not? How many times did the participants evaluate?

What do you mean “The funder played no role in the design, conduct, or reporting of this study.” It is written that the manuscript has no funding resources.

Statistical analysis and results

Do you have any special reason for comparing median age, height, and weight? It seems a sufficient number to compare with t-test.

Are there any age-specific characteristics in the pattern of MVPA levels? I wonder if they had a different percentage of sufficient MVPA level according to the age group. With the supplementary data, I realized that a relatively small number of 6-aged students and they all are almost in suboptimal PA group, for example.

It would be better to demonstrate a general logistic regression table instead of a simple P-value and 95% CI.

Discussion

Recommend some of the points to discuss.

One study revealed that disability significantly affects muscle mass. (Pediatr Int. 2020 Aug;62(8):962-969. doi: 10.1111/ped.14248.)

Another study figured out that PA is also associated with positive mental health in a multi-ethnic Asian population. (Int J Environ Res Public Health. 2020 Nov 16;17(22):8489. doi: 10.3390/ijerph17228489.)

Reviewer #2: Overall Impressions: Well done study. Statistical analysis appears appropriate. Sample size appears appropriate. Some limitations in methodology need to be addressed.

Specific concerns:

a) Please report on the reliability and precision of muscle function and physical performance tests. Should provide some evidence of this to ensure tests were reliable.

b) Use of BIA to assess appendicular skeletal muscle mass is impacted by normative equations used by equipment. Authors need to address the limitations of BIA in the discussion and to report on precision and reliability data.

c) Use of subjective measures of PA (questionnaire) is limited and may have impacted overall study findings. It would be best to explore the extent to which bias could have impacted overall study findings in discussion. This should be expanded in more detail in the discussion as this would have limited overall study findings.

d) Making the assumption of physical activity outside of ‘school hours’ may not be appropriate and hence the authors should temper their overall conclusions on this point.

e) Any seasonal variations in physical activity noted? This would be important to include as school aged children PA patterns may vary substantively over the winter vs summer/spring months.

6. PLOS authors have the option to publish the peer review history of their article (what does this mean?). If published, this will include your full peer review and any attached files.

Reviewer #1: No

Reviewer #2: No

---

## [Author Response · Author response to Decision Letter 0]

12 Feb 2021

Editor #1

Comments

Journal Requirements:

Response: The authors would like to thank the Editor for the constructive critique to improve the manuscript. We have made every effort to address the issues raised and to respond to all comments. The revisions are indicated in red font in the revised manuscript. 

Comments

'The funders had no role in study design, data collection and analysis, decision to publish, or preparation of the manuscript.'

Please clarify the sources of funding (financial or material support) for your study. List the grants or organizations that supported your study, including funding received from your institution.

State what role the funders took in the study. If the funders had no role in your study, please state: “The funders had no role in study design, data collection and analysis, decision to publish, or preparation of the manuscript.”

If any authors received a salary from any of your funders, please state which authors and which funders.

If you did not receive any funding for this study, please state: “The authors received no specific funding for this work.”

Response: We would like to thank the Editor for the comment. Please note that we have not received any specific funding for this work. This information has been included and clarified in the cover letter and manuscript files in accordance with the Editor’s comment.

Reviewer #1

The authors would like to thank the reviewer for his/her constructive critique to improve the manuscript. We have made every effort to address the issues raised and to respond to all comments. The revisions are indicated in red font in the revised manuscript. Please, find next a detailed, point-by-point response to the reviewer's comments.

Comment

Thank you for the hard-effort observational study. This study has tried to reveal the relationship between sufficient PA and skeletal muscle mass in growing children. And even more, authors tried to figure out the real physical performance by examining several time-consuming studies such as TUGT, one-leg standing time, sit to stand test results, grip strength, gait, etc.

Response: We would like to thank the reviewer for the positive evaluation of our work.

Comment

Method

Please define when skeletal muscle mass assessment with BIA was done as the previous food, or liquid consumption could influence the results.

Response: We would like to thank the reviewer for the constructive comment. Please note that we have added this information to the sentence describing the skeletal muscle mass assessment with BIA, as per the reviewer’s suggestion.

“Bioelectrical impedance analysis (BIA) was only performed at 2 h after the meals, as described in the instruction manual.” (Lines 145–147)

Comment

Would you please describe details about the grip strength? Did you have a practice trial or not? How many times did the participants evaluate?

Response: We would like to thank the reviewer for pointing this out. Grip strength was measured once each to the right and left limbs. Moreover, the participants were allowed to perform one trial, as a learning attempt. We have amended this information in the Methods section, as follows: “The dominant hand was measured first, followed by the non-dominant hand. The grip strength of both hands was measured once each with the participants in the sitting position with their shoulder abducted and neutrally rotated, the elbow at extension, and the forearm and wrist in the neutral position [28]. The participants were instructed to squeeze the handle of the dynamometer as hard as they could and sustain for 5 s [29]. Verbal encouragement (i.e., “Keep it up, keep it up”) was provided during the test [29]. Prior to the grip strength assessment by an experienced physical therapist, the participants were allowed to perform one practice trial as a learning attempt. The grip strength measurement reliability was reported to be excellent in a full elbow-extension position [30].” (Lines 173–181)

Comment

What do you mean “The funder played no role in the design, conduct, or reporting of this study.” It is written that the manuscript has no funding resources.

Response: We would like to thank the reviewer for the comment. Please note that we have received any specific funding for this work. This error has been corrected in accordance with the reviewer’s comment.

Comment

Statistical analysis and results

Do you have any special reason for comparing median age, height, and weight? It seems a sufficient number to compare with t-test.

Response: We would like to thank the reviewer for providing these insightful comments. The t-test cannot be performed because the age, height, and weight data were not normally distributed.

Comment

Are there any age-specific characteristics in the pattern of MVPA levels? I wonder if they had a different percentage of sufficient MVPA level according to the age group. With the supplementary data, I realized that a relatively small number of 6-aged students and they all are almost in suboptimal PA group, for example.

Response: We would like to thank the reviewer for pointing this out. The percentages of children that performed the recommended MVPA level according to their age groups were as follows: 6-year-old, 11.8%; 7-year-old, 25.0%; 8-year-old, 31.6%; 9-year-old, 37.7%; 10-year-old, 57.6%; 11-year-old, 72.3%; and 12-year-old, 53.0%. With respect to this study characteristic, we presumed that children aged 10–12 actively participated in school clubs, and the opportunities for participation in sports activities are numerous. We have provided this information in the Results section. 

Comments

It would be better to demonstrate a general logistic regression table instead of a simple P-value and 95% CI.

Response: We would like to thank the reviewer for the comment. We agree with the reviewer’s suggestion. Therefore, we have included this information in Table 3.

Comment

Discussion

Recommend some of the points to discuss.

One study revealed that disability significantly affects muscle mass. (Pediatr Int. 2020 Aug;62(8):962-969. doi: 10.1111/ped.14248.)

Another study figured out that PA is also associated with positive mental health in a multi-ethnic Asian population. (Int J Environ Res Public Health. 2020 Nov 16;17(22):8489. doi: 10.3390/ijerph17228489.)

Response: We would like to thank the reviewer for the suggestion. We agree that the points mentioned need to be added to the Discussion section. Following the reviewer's insightful suggestion, we have revised these sentences in the Discussion section and cited the relevant literature, as follows:

“Moreover, one study revealed that disability significantly affected the muscle mass [47].” (Lines 313–314)

“Another study reported that PA was also associated with positive mental health in a multi-ethnic Asian population [65]. We presumed that children in our study actively participated in school clubs, and the opportunities for participation in sports activities are numerous.” (Lines 358–361)

Reviewer #2

We would like to thank the reviewer for providing insightful comments regarding our manuscript. We express our sincere gratitude for the time and effort the reviewer has spent to review our manuscript. The comments and suggestions have helped us immensely improve our work.

Comments

Overall Impressions: Well done study. Statistical analysis appears appropriate. Sample size appears appropriate. Some limitations in methodology need to be addressed.

Response: We would like to thank the reviewer for the positive comments regarding our study. The manuscript has been rechecked, and the necessary changes have been made in accordance with the reviewers’ suggestions. The responses to all comments have been prepared and given below.

Comments

a) Please report on the reliability and precision of muscle function and physical performance tests. Should provide some evidence of this to ensure tests were reliable.

Response: We agree with the reviewer’s comment. Therefore, we have provided this information in the Materials and Methods section (Assessment of appendicular skeletal muscle mass, Grip strength, Five times sit-to-stand test, Timed Up-and-Go test, One-leg standing time, and Gait efficiency).

Comments

b) Use of BIA to assess appendicular skeletal muscle mass is impacted by normative equations used by equipment. Authors need to address the limitations of BIA in the discussion and to report on precision and reliability data.

Response: Following the reviewer’s suggestion, we have added more information to the Discussion section as follows: “Fourth, the validity of the BIA method depends on the participant’s hydration status. Limitations of BIA include assumptions involving a fixed hydration status [67]. Thus, BIA may not always be the best method to assess body composition.” (Lines 375–377)

Comments

c) Use of subjective measures of PA (questionnaire) is limited and may have impacted overall study findings. It would be best to explore the extent to which bias could have impacted overall study findings in discussion. This should be expanded in more detail in the discussion as this would have limited overall study findings.

Response: We would like to thank the reviewer for pointing this out. We agree with the reviewer’s comment. Therefore, we have added the following part to the Discussion section: 

" However, the categorization of MVPA relied on self-reported questionnaire data that could be affected by recall bias. Subjective assessment was reported to overestimate MVPA compared with objective assessment [66]. Moreover, as the items for measuring preference for MVPA were not investigated for reliability and validity, a future longitudinal study is needed to confirm this hypothesis.” (Lines 364–368)

Comments

d) Making the assumption of physical activity outside of ‘school hours’ may not be appropriate and hence the authors should temper their overall conclusions on this point.

Response: We agree that the assumption of physical activity outside of “school hours” needs improvement. Therefore, we have revised the corresponding part in the manuscript to avoid its misinterpretation as follows: “We presumed that children in our study actively participated in school clubs, and the opportunities for participation in sports activities are numerous.” (Lines 360–361)

Comments

e) Any seasonal variations in physical activity noted? This would be important to include as school aged children PA patterns may vary substantively over the winter vs summer/spring months.

Response: We would like to thank the reviewer for the suggestion. Differences between the four seasons and the physical activity time were examined using one-way analysis of the Kruskal–Wallis test. However, the physical activity levels were not significantly different because of the seasonal variations in this study (P=0.061; η2=0.02), indicating that these seasonal variations observed in the winter vs. summer/spring months did not necessarily exhibit significant changes in physical activity time. We agree that we should focus on the seasonal variations and, therefore, we have provided more information in S2 Table.

---

## [Decision Letter · Decision Letter 1]

12 Mar 2021

PONE-D-20-38059R1

Relationship between skeletal muscle mass index and physical activity of Japanese children: A cross-sectional, observational study

PLOS ONE

Dear Dr. Ito,

Thank you for submitting your manuscript to PLOS ONE. After careful consideration, we feel that it has merit but does not fully meet PLOS ONE’s publication criteria as it currently stands. Therefore, we invite you to submit a revised version of the manuscript that addresses the points raised during the review process.

The manuscript is still necessary to be revised before acceptance.

See carefully the Reviewers' comments and respond them appropriately.

We look forward to receiving your revised manuscript.

Kind regards,

Masaki Mogi

Academic Editor

PLOS ONE

Reviewers' comments:

Reviewer's Responses to Questions

**Comments to the Author**

1. If the authors have adequately addressed your comments raised in a previous round of review and you feel that this manuscript is now acceptable for publication, you may indicate that here to bypass the “Comments to the Author” section, enter your conflict of interest statement in the “Confidential to Editor” section, and submit your "Accept" recommendation.

Reviewer #3: All comments have been addressed

Reviewer #4: (No Response)

2. Is the manuscript technically sound, and do the data support the conclusions?

Reviewer #3: Yes

Reviewer #4: Partly

3. Has the statistical analysis been performed appropriately and rigorously? 

Reviewer #3: Yes

Reviewer #4: Yes

4. Have the authors made all data underlying the findings in their manuscript fully available?

Reviewer #3: Yes

Reviewer #4: Yes

5. Is the manuscript presented in an intelligible fashion and written in standard English?

Reviewer #3: Yes

Reviewer #4: No

6. Review Comments to the Author

Reviewer #3: - Please define in the introduction "physical function" and which variables comprise it. Include what studies show evidence of it in this population.

- They should better justify why they have chosen these variables "gait speed", "Five times sit-to-stand test" and "Timed Up-and-Go test" in this population. They are validated tests and recommended mainly for older population. Since their study is not with children with cerebral palsy as the articles that they put as reference.

- LINE 342: As indicated, the test results do not correlate with handgrip strength. This is because the test is not the most appropriate for this population. Just a comment to be taken into account in future research.

If these considerations are included, the article is suitable for publication.

Reviewer #4: The subject is interesting and the study is well designed. However, the following corrections must be done before further considerations.

- The abstract should be revised in terms of scientific writing. For example, the first sentence is short and vague. Which confounders were adjusted in regression models?

-Introduction needs to be revised. It should be started with describing the health issue (body composition in adolescence) not the independent variable.

- Explain about school selection criteria.

- Some drugs may have effects on weight and body composition. Did you consider using them as the exclusion criteria? If not, suggested to mention it in your limitations.

- Add decryptions about you bio impedance analyzer.

- The role of dietary intake in determining body composition was ignored. Mention it as a limitation if you did not collect data on diet.

- Discuss about the possible effects of genetics and diet on body composition and also their effects on the association between physical activity and body muscle mass.

See these papers: https://doi.org/10.1016/j.clnu.2018.06.1124, https://dx.doi.org/10.1080%2F21623945.2019.1693745, https://dx.doi.org/10.1186%2Fs12967-019-1921-4.

- Conclusion must be drawn from your main results and should be revised.

7. PLOS authors have the option to publish the peer review history of their article (what does this mean?). If published, this will include your full peer review and any attached files.

Reviewer #3: No

Reviewer #4: No

---

## [Author Response · Author response to Decision Letter 1]

5 Apr 2021

Reviewer #3

The authors would like to thank the reviewer for his/her constructive critique to improve the manuscript. We have made every effort to address the issues raised and to respond to all the comments. The revisions are indicated in red font in the revised manuscript. Please, find our detailed, point-by-point responses to the reviewer's comments below.

Comment

Please define in the introduction "physical function" and which variables comprise it. Include what studies show evidence of it in this population.

Response: Thank you for pointing this out. We agree with the reviewer’s suggestion. We have added this information and cited the relevant literature in the Introduction as requested. (Lines 94–105)

Comment

They should better justify why they have chosen these variables "gait speed", "Five times sit-to-stand test" and "Timed Up-and-Go test" in this population. They are validated tests and recommended mainly for older population. Since their study is not with children with cerebral palsy as the articles that they put as reference.

Response: We agree it is very important to justify for the variables used in this study by citing previous work. The gait speed is a critical element of physical function and has been reported to serve as a fundamental indicator for gait development in children [14-16]. A previous study reported that the FTSST and TUG are feasible and reliable tools for both children with typical development and those with disabilities [18-22]. We have added this information in the introduction section. (Lines 96–105).

Comment

LINE 342: As indicated, the test results do not correlate with handgrip strength. This is because the test is not the most appropriate for this population. Just a comment to be taken into account in future research.

Response: We would like to thank the reviewer for the constructive comment. Please note that we have added this information to the sentence describing results of the grip strength, as per the reviewer’s suggestion.

“Therefore, the grip strength test was found to not be the most appropriate for this population. (Line 384)

Reviewer #4

We express our sincere gratitude for the time and effort the reviewer has spent reviewing our manuscript and for providing insightful comments and suggestions, which have helped us immensely to improve our work.

Comments

The subject is interesting and the study is well designed. However, the following corrections must be done before further considerations.

Response: Thank you for raising these issues. The manuscript has been rechecked carefully, and the necessary changes have been made in accordance with the reviewers’ suggestions. The responses to all comments have been prepared and are provided below.

Comments

The abstract should be revised in terms of scientific writing. For example, the first sentence is short and vague. Which confounders were adjusted in regression models?

Response: We agree with the reviewer’s comment. Following the reviewer’s suggestion, we have added the following additional information to the abstract section: “Regular physical activity is an important component of physical health of children and has been associated with increasing skeletal muscle mass and muscle strength. Children with low levels of physical activity may experience health problems, such as loss of muscle mass, later in life. Thus, it may be valuable to identify declining physical function in children who do not perform the recommended amount of physical activity. Therefore, we aimed to evaluate the relationship between the amount of physical activity performed for ≥60 min per day for ≥5 days per week and the skeletal muscle mass index and physical function in young children.” and “Multiple logistic regression analyses were performed to assess the association of moderate-to-vigorous physical activity with the skeletal muscle mass index, percent body fat, and physical function, after controlling for confounding factors (age and sex).” (Lines 50–57 and 62–65)

Comments

Introduction needs to be revised. It should be started with describing the health issue (body composition in adolescence) not the independent variable.

Response: We would like to thank the reviewer for pointing this out and agree with the reviewer’s comment. Thus, we have added the following details to the Introduction section: “A physically active lifestyle improves muscle health or physical health and has a positive effect on body composition and physical function in childhood and adolescence [1].” (Lines 74–75)

Comments

Explain about school selection criteria.

Response: We thank the reviewer for pointing this out and agree with the reviewer’s comment. Thus, we have added the following information to the Study population subsection: "Of the 48 elementary schools in Okazaki City, two schools introduced to us by the Okazaki City Board of Education, from which we received informed consent for study participation, were included." (Lines 120–122)

Comments

Some drugs may have effects on weight and body composition. Did you consider using them as the exclusion criteria? If not, suggested to mention it in your limitations.

Response: We agree with the reviewer’s comment. Therefore, we have added the following details to the Study population: "Of note, six of the excluded participants were being treated with medications that could influence body composition and body weight. None of the other participants were receiving such medications." (Lines 133–135).

Comments

Add decryptions about you bio impedance analyzer.

Response: Accordingly, we have added the following details to the “Assessment of appendicular skeletal muscle mass” section: "The multi-frequency bioelectrical impedance analyzer is used to determine the relationship between the measured volume of an electrical resistance and the conductor. The analyzer uses multiple frequencies to differentiate intracellular fluid from extracellular fluid and, therefore, provides an estimation of total body water [34]. Fat and bone have correspondingly high impedance and low conductivity, and skeletal muscle is an electrolyte rich tissue with low resistance." (Lines 165–170)

Comments

The role of dietary intake in determining body composition was ignored. Mention it as a limitation if you did not collect data on diet.

Response: Thank you for pointing this out. We had previously mentioned that we could not account for effects of dietary intake on the overall body composition in the study limitations in the original manuscript (Line 377), but in this revised version we have added the following to the limitation section to avoid any potential confusion: "we could not account for the role of dietary intake…" (Lines 420–421).

Comments

Discuss about the possible effects of genetics and diet on body composition and also their effects on the association between physical activity and body muscle mass.

See these papers: https://doi.org/10.1016/j.clnu.2018.06.1124, https://dx.doi.org/10.1080%2F21623945.2019.1693745, https://dx.doi.org/10.1186%2Fs12967-019-1921-4.

Response: Following the reviewer’s suggestion, we have discussed this issue in the revised manuscript, and we have cited the suggested papers. We have added the following to the Discussion: "The findings of Doaei et al. [65,66] indicated that the FTO gene may play a vital role in the increase in skeletal muscle mass and body composition in male adolescents. Furthermore, lifestyle modifications may exert their effects on obesity through changes in the expression of the FTO and IRX3 genes [65]. Moreover, lifestyle modification appeared to have an impact on obesity through changes in the expression of the FTO and IRX3 genes [67]." (Lines 354–359)

Comments

Conclusion must be drawn from your main results and should be revised.

Response: Thank you for pointing this out. This error has been corrected in accordance with your comment. (Lines 427–428)

---

## [Decision Letter · Decision Letter 2]

19 Apr 2021

Relationship between skeletal muscle mass index and physical activity of Japanese children: A cross-sectional, observational study

PONE-D-20-38059R2

Dear Dr. Ito,

We’re pleased to inform you that your manuscript has been judged scientifically suitable for publication and will be formally accepted for publication once it meets all outstanding technical requirements.

Kind regards,

Masaki Mogi

Academic Editor

PLOS ONE

Additional Editor Comments (optional):

No further comment.

Reviewers' comments:

Reviewer's Responses to Questions

**Comments to the Author**

1. If the authors have adequately addressed your comments raised in a previous round of review and you feel that this manuscript is now acceptable for publication, you may indicate that here to bypass the “Comments to the Author” section, enter your conflict of interest statement in the “Confidential to Editor” section, and submit your "Accept" recommendation.

Reviewer #3: All comments have been addressed

Reviewer #4: All comments have been addressed

2. Is the manuscript technically sound, and do the data support the conclusions?

Reviewer #3: (No Response)

Reviewer #4: Yes

3. Has the statistical analysis been performed appropriately and rigorously? 

Reviewer #3: Yes

Reviewer #4: Yes

4. Have the authors made all data underlying the findings in their manuscript fully available?

Reviewer #3: Yes

Reviewer #4: Yes

5. Is the manuscript presented in an intelligible fashion and written in standard English?

Reviewer #3: Yes

Reviewer #4: Yes

6. Review Comments to the Author

Reviewer #3: Dear authors

The article is very interesting and the authors have made the requested changes and the article can be accepted for publication.

Congratulations

Best regards

Reviewer #4: This paper aimed to investigate the association between skeletal muscle mass index and physical activity of Japanese children. All comments have been addressed.

7. PLOS authors have the option to publish the peer review history of their article (what does this mean?). If published, this will include your full peer review and any attached files.

Reviewer #3: No

Reviewer #4: No

---

## [Editor Report · Acceptance letter]

27 Apr 2021

PONE-D-20-38059R2 

Relationship between the skeletal muscle mass index and physical activity of Japanese children: A cross-sectional, observational study 

Dear Dr. Ito:

I'm pleased to inform you that your manuscript has been deemed suitable for publication in PLOS ONE. Congratulations! Your manuscript is now with our production department. 

Kind regards, 

on behalf of

Dr. Masaki Mogi 

Academic Editor

PLOS ONE